# Association of social vulnerability factors with power outage burden in Washington state: 2018–2021

Claire A. Richards[1]*, Solmaz Amiri[2], Von P. Walden[3], Julie Postma[1], Mohammad Heidari Kapourchali[4], Alain F. Zuur[5]

1 Department of Nursing and Systems Science, College of Nursing Washington State University College of Nursing, Spokane, WA, United States of America, 2 Elon S. Floyd College of Medicine, Washington State University, Spokane, WA, United States of America, 3 Department of Civil and Environmental Engineering, Washington State University, Pullman, WA, United States of America, 4 Department of Electrical Engineering, University of Alaska Anchorage, Anchorage, Alaska, United States of America, 5 Highland Statistics Ltd., Newburgh, United Kingdom

* claire.richards@wsu.edu

**Data Availability Statement:** All processed data and code used for data exploration, model fitting, creating tables and plotting is available on a figshare repository at https://doi.org/10.6084/m9.

## Abstract

Major power outages have risen over the last two decades, largely due to more extreme weather conditions. However, there is a lack of knowledge on the distribution of power outages and its relationship to social vulnerability and co-occurring hazards. We examined the associations between localized outages and social vulnerability factors (demographic characteristics), controlling for environmental factors (weather), in Washington State between 2018–2021. We additionally analyzed the validity of PowerOutage.us data compared to federal datasets. The population included 27 counties served by 14 electric utilities. We developed a continuous measure of daily outage burden using PowerOutage.us data and operationalized social vulnerability using four factors: poverty level, unemployment, disability, and limited English proficiency. We applied zero-altered lognormal generalized additive mixed-effects models to characterize the relationship between social vulnerability and daily power outage burden, controlling for daily minimum temperature, maximum wind speed, and precipitation, from 2018 to 2021 in Washington State. We found that social vulnerability factors have non-linear relationships with outages. Wind and precipitation are consistent drivers of outage occurrence and duration. There are seasonal effects that vary by county-utility area. Both PowerOutage.us and federal datasets have missing and inaccurate outage data. This is the first study evaluating differential exposure to localized outages as related to social vulnerability that has accounted for weather and temporal correlation. There is a lack of transparency into power outage distribution for those most vulnerable to climate impacts, despite known contributions by electric utilities to climate change. For effective public health surveillance of power outages and transparency, outage data should be made available at finer spatial resolution and temporal scales and/or utilities should be required to report differential exposure to power outages for socially vulnerable populations.

figshare.24908559. Raw, unprocessed data can be purchased from PowerOutage.us.

**Funding:** Washington State University New Faculty Seed Grant [PG00019865]. The funders had no role in study design, data collection and analysis, decision to publish, or preparation of the manuscript.

**Competing interests:** The authors have declared that no competing interests exist.

## Introduction

Major power outages (POs) have risen recently in the United States because of insufficient investment in aging infrastructure [1] and more frequent and severe extreme weather events due to climate breakdown [2–4]. In the last 20 years, 46%-53% of major POs were related to severe weather [2,5] and over the last decade, weather-related outages have increased by 78% [6]. POs pose public health risks due to the disruption of temperature regulation, refrigeration, air purification, water pumps, emergency response, communication systems, and the use of medical equipment [7–10]. Documented PO impacts include increased all-cause mortality and morbidity, respiratory, cardiovascular, and renal disease hospitalizations, and pregnancy complications [7,8,11,12]. Recent studies have shown that socially vulnerable populations experience longer and more extensive POs [13–19]. This is especially concerning because these same populations often possess fewer financial and institutional resources to cope with POs [20].

The power industry has traditionally focused on the role of energy infrastructure in vulnerability to POs, but there is growing interest in the role of social vulnerability in the exposure to and impact of POs [14,21]. In disaster management, social vulnerability is conceptualized as a multidimensional process that emphasizes the role of social, institutional, political and economic systems that shape future experiences of disasters [22–25]. These processes result in disadvantages for some groups and advantages for others [23]. Knowledge of the link between social vulnerability and POs could provide energy regulators, electric utilities, and emergency managers with evidence needed to apply equity and justice concerns in planning and decision-making. However, a lack of consensus and interpretability of PO measures pose challenges to their application.

Most ecological studies of the health and social impacts of POs have focused on large-scale events such as the Northeast Blackout of 2003 [26–29], Winter Storm Uri of 2021 [15,17], and numerous hurricanes [11,18,30,31]. The substantial number of studies on major events indicates interest in understanding resilience, including power infrastructure, health system, and community resilience in the context of escalating climate change. The Institute of Electrical and Electronics Engineers (IEEE) has developed guidelines for identifying major events and separating them from routine reliability metrics [32]. Thresholds for major events are calculated based on the normal operation for each electric utility by identifying statistical outliers in the distribution of daily natural log-transformed System Average Interruption Duration Index (SAIDI) values [32]. When the overall reliability of a utility declines, the threshold for major events increases. As a result, definitions distinguishing between major and non-major outages are based on statistical distributions and are inconsistent. Further, they do not identify or define non-major events (localized POs that are not widespread) of public health significance, overlooking moderate or even small POs that pose serious health threats or hardships for socially vulnerable residents.

There have been some studies on the health impacts or exposure to localized POs, or smaller scale POs [7,11,12,14,33]. These studies identified POs through a daily median threshold rather than the start of a major event, ranging from 0.37% to 2.2% of affected customers, based on the distribution of daily PO coverage [7,11,12]. The intensity of these POs were then defined according to the quantile of PO coverage and the number of consecutive days [7,11,12]. There is a lack of evidence for defining localized POs according to statistical distributions, however, and this practice might lead to spurious findings and difficulty comparing results across studies. Furthermore, most studies of localized POs have been conducted in New York State due to the availability of data provided by the Department of Public Service [7,11,12,34]. None of these studies have described differences in exposure to localized POs according to social vulnerability factors.

In a nationwide study of localized POs and social vulnerability between 2018–2020 using PowerOutage.us data, Do et al. defined a medically-relevant PO as 0.1% of customers affected for 8 hours, and based the threshold on the 90[th] percentile out per hour [13]. Like PO studies conducted in New York State, the validity of such a threshold remains unclear. The authors reported that counties in the highest quartile of Social Vulnerability Index (SVI) experienced more medically-relevant outages. Conversely, counties in the highest quartile of durable medical equipment (DME) use among Medicare beneficiaries had fewer such POs than other counties. Notably, disability increases with DME use and is included as a component of the SVI [13], underscoring the difficulty of interpreting overall SVI scores. Furthermore, PowerOutage.us data has been used recently in several ecological studies [13,15,16,30], it has not been validated or compared with other data sources.

In this study, we examine the relationship between social vulnerability factors and county-level outage burden across Washington State between 2018 and 2021, controlling for weather variables, including wind, rain, and temperature. We chose to use the daily SAIDI value as a continuous metric of county-level outage burden because it is a standardized metric in the power industry often reported annually and allows for the validation of outage metrics by comparison with established federal datasets [35,36]. Defined as the average outage duration among customers served, SAIDI is a continuous metric that integrates both duration and scale of POs. As a continuous metric, SAIDI possesses more information and is more sensitive to changes than categorical measures [37,38]. We used a daily measure of SAIDI to incorporate daily weather variability and to address the limitations of missing data. Our secondary objective is to assess the validity of the PowerOutage.us data by calculating utility and state-wide annual SAIDI estimates, identifying and describing the characteristics of major PO events, and comparing our results with established federal datasets.

## Materials and methods

### Power outage data and study population

Sustained PO data was obtained from PowerOutage.us, a platform that collects, records, and aggregates live PO data. This information, gathered through an application programming interface (API), includes data from utilities that provide web portals displaying POs for their customers [39]. The PowerOutage.us data are comprised of rows of dates and times in the UTC time zone and includes variables for the utility, state, county, subdivision, customers tracked, customers out, and date-time (S1 Table). PowerOutage.us minimizes storage requirements by only storing a date-time stamp when the number of customers changes. The PowerOutage.us checks the utility API every 10–20 minutes for changes in the stored values, according to email correspondence [40]. Utilities aggregate outage data at varying geographic scales—some report exclusively at the county level ($n = 7$), while others provide data for subdivisions ($n = 15$); additionally, this reporting varies over time (S1 Fig). The subdivision variable names are given by utilities for the purposes of operating their website and often do not correspond with geography. The PowerOutage.us variable for customers tracked does not reliably represent the number of customers served by the utility in the geographic area. Our study did not require institutional review board review or approval because it does not involve human subjects.

We aggregated the PO data on the county level for each electric utility (county-utility) rather than on the subdivision level due to missing and inconsistent subdivision information. We also aggregated PO data on the county-utility level rather than on the county level due to missing data for different utilities within the same county. We excluded electric utilities with

only partial data and/or low data quality (Fig 1) after conducting extensive pre-processing of the data (S1 File and S2 Table).

The primary analysis included six of 65 electric utilities serving 20 counties in Washington State from February 17, 2018 to December 31, 2021. These six utilities served 2.6 million customers in 23 county-utility service areas and reported outages on 94% of the study days. A secondary analysis added eight utilities with a lower percentage of observations suggesting lower data reliability. The secondary analysis included a total of 14 utilities serving 3.0 million customers in 27 counties ($n = 31$ county-utilities). For confidentiality, unique identifiers were assigned to each electric utility in Washington State.

## County-utility customer counts

To estimate daily SAIDI, customer counts (metered service points, not people or households) for each county-utility area were needed. However, only data for state-wide customer counts were available from the U.S. Energy Information Administration (EIA). The state-wide counts were equivalent to county-utility counts when the utility only operates in a single county, but many utilities operate in more than one county [41].

To estimate all Washington utility counts, we estimated their state-wide residential, commercial, and industrial customer counts using the Forms EIA-861 and EIA-861S [13,36]. To

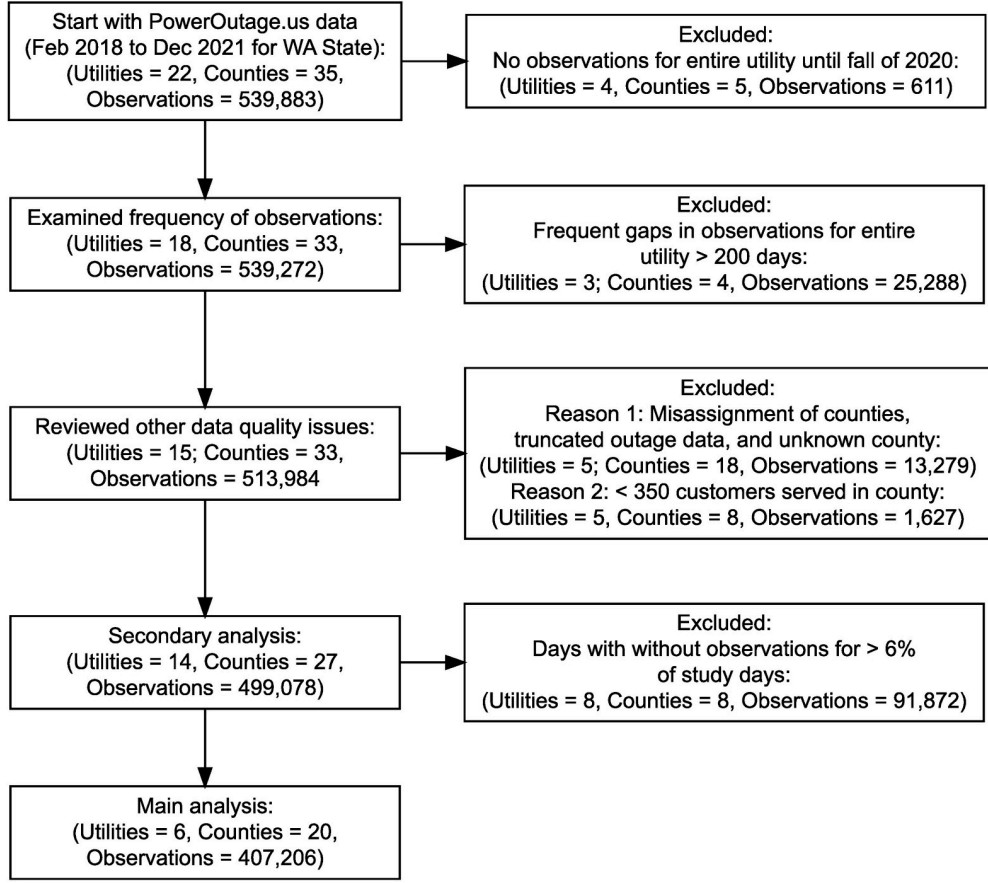

**Fig 1. Flow chart for the selection of county-utility service areas.** Additional zero observations removed from primary and secondary analysis as described in S1 Table. Raw unprocessed PowerOutage.us data included 22 utilities.

determine the fraction of customers served by utilities in each county for each study year, we first contacted electric utilities to request county-level customer counts and extracted information from utility websites. We then estimated the fraction of customers served by utilities in each county. For consistency, we estimated the county-utility customer counts for each year by multiplying the utilities total state-wide counts from EIA by the fraction of customers served for each county.

In prior research, analyses have been conducted on the level of the county [13]. Do et al. employed a downscaling method to estimate county customer counts [13]. This approach involved apportioning the total state-wide customer counts to individual counties based on each county's proportion of households and establishments. As an ancillary analysis, we leveraged utility data we had collected to compare our estimates with these downscaling methods. To replicate the downscaling methods, we derived the number of households from the 2017–2021 American Community Survey (ACS), and the number of establishments per county from the Census Business Patterns data for each year of our study [42]. We quantified the error associated with downscaling methods by comparing these with the aggregated utility-derived data for each county, thereby informing future research approaches.

## Outage burden and major events

Our measure of outage burden is the daily SAIDI for county-utility areas. We calculated SAIDI by dividing the sum of the customer-outage time (number of customers experiencing an outage multiplied by the duration of the outage) for each county-utility and day by the number of county-utility customers on the system for the year [43]. We initially explored other metrics such as Customer Average Interruption Duration Index (CAIDI) and System Average Interruption Frequency Index (SAIFI), but these metrics were more affected by spurious zero values than SAIDI because they require the identification of outage events (S1 and S2 Tables).

As part of validating the PowerOutage.us data, we compared our utility- and state-wide SAIDI results with EIA data [36]. The EIA-861 includes data on annual reliability information including the SAIDI with and without major events (defined as days when the daily system SAIDI exceeds a threshold value). Utilities may calculate the reliability metrics using either the Institute of Electrical and Electronics Engineers (IEEE) 1366–2012 [43] or IEEE 1366–2005 or choose not to certify to an IEEE standard.

We additionally defined and described major outage events in both absolute and relative terms [15]. Absolute definitions allow for the accounting of the PO magnitude and comparison with Department of Energy (DOE) definition of major PO events [35], whereas relative definitions allow for the identification of a similar number of PO events across counties with different population sizes. In absolute terms, major events were defined as days with at least 10,000 and 50,000 customer-POs affected in an hour [15]. Relative major events were defined as: 1) PO days affecting 0.1% of county-utility customers for eight consecutive hours, 2) a major event day (MED) with SAIDI exceeding the threshold ($T_{MED}$), defined as the exponential of the sum of $\alpha$ and 2.5 times $\beta$ [43]. Here, $\alpha$ is defined as the mean of the natural logarithm of all non-zero daily county-utility SAIDI and $\beta$ is the standard deviation about that mean [43]. We compared our major events with those reported on the DOE-417, "Electric Emergency Incident and Disturbance Report" [35].

## Social vulnerability

We operationalized social vulnerability using individual factors rather than summary scores or themes from social vulnerability indices used by other research [13,15,31]. We chose individual factors rather than summary scores for two main reasons: firstly, individual factors offer

more specific, actionable insights for targeted interventions; and secondly, this approach addresses concerns about the validity of the most commonly used tools available on the county level [25,44,45]. We initially considered 10 social vulnerability factors but dropped six variables due to collinearity and variance inflation factors (VIFs) (households occupying multi-unit housing, percentage of population: with reliance on electricity-dependent medical equipment, aged 65 years and older living alone, aged 5 years and younger, non-white and non-Hispanic, and percentage of households living in mobile homes). We additionally considered household density and rurality (population living in an urban vs. rural area) as potential control measures for grid density, but these were also dropped due to collinearity with social vulnerability factors. We conducted all analyses with the four highest priority factors due to their theoretical relationship with vulnerability to POs [15,18,20,21,30], including percentage within the county-utility service territory of: 1) households living under 100% of the federal poverty limit, 2) civilian population 18 years of age or older with a disability, 3) households with non-English language preference ("limited English"), square root-transformed due to skewness in the data, and 4) population unemployed. We standardized all continuous social vulnerability factors with Z-score standardization in our generalized linear mixed models (GLMMs).

We started with the Electric Retail Service Territories map developed by the Oak Ridge National Laboratory (ORNL), revising utility boundaries based on information provided by utility websites [46]. We ascertained county-utility level demographic information from the American Community Survey (ACS), 2017–2021 (5-year) data and retrieved from the National Historical Geographic Information System (NHGIS) [47]. To do this, we used population-weighted centroids for census block groups to represent the locations of populations from the 2020 Census [47,48]. We then overlayed the population-weighted block group centroids with the service territories and allocated the populations from the ACS data to each service territory using QGIS v3.30.1 [49].

## Weather variables

Hourly temperature and wind data were obtained from the High-Resolution Rapid-Refresh (HRRR) model [50,51], a weather forecasting model produced by the U.S. National Weather Service. We chose to use the HRRR analysis data because it provides a good representation of weather events over the Pacific Northwest, and it has a fine spatial resolution that resolves the complex topography of Washington State (Olympic and Cascade Mountain ranges). Data from the HRRR analysis fields were used, which assimilate real-time data from a variety of sources including surface observations, regional weather networks, radar data, and satellite products. Observations are assimilated into HRRR analyses for each hourly forecast at a spatial resolution of 3 km using the Gridpoint Statistical Interpolation system, which provides hourly values of surface temperature, humidity, and horizontal wind. We used hourly 2-m air temperature (TMP; 2m_above_ground) to determine the minimum and maximum 2-m air temperatures (˚C) and the hourly 10-m maximum wind speed (WIND_max_fcst; 10m_above_ground) to daily maximum wind speed (m/s) for each day of the study period.

In addition, we used gridMET data for precipitation accumulation on each calendar day from midnight-midnight local time [52]. GridMET is a hybrid dataset that combines spatially downscaled weather data from the North American Land Data Assimilation System (NLDAS) with date from the Parameter-elevation Relationships on Independent Slopes Model (PRISM) [52].

## Statistical analysis

We conducted data exploration following the protocol described in Zuur et al. [53]. The response variable was the average duration of POs per day (SAIDI) in minutes. We included the social vulnerability metrics and weather variables as predictors.

**Distribution.** We applied a zero-altered lognormal (ZALN) model within the context of generalized additive mixed effects model (GAMM) using the bam() function from "mgcv" [54]. In such a model, the absence-presence data is analyzed with a Bernoulli model and the non-zero data are analyzed with a log-normal model [55]. The choice for using a log-normal distribution for the non-zero data was partly motivated by the fact that it enhanced numerical stability for our advanced models applied to large data sets, ensuring more reliable convergence and accuracy in our estimations.

The analysis incorporated social vulnerability factors and weather variables. A GAMM was used to allow for non-linear covariate effects, providing flexibility in modeling complex relationships between predictors and the response variable [54,56]. Cubic regression splines were utilized for the smoothers. We used fast REML to estimate the smoothing parameters and illustrated the plots with "gratia" [57] and "ggplot2" [58] packages. All analyses were conducted in R 4.3.1 [59].

**Dependency.** To avoid pseudo-replication, we included random effects and modeled the temporal patterns using smoothing functions of time. To capture the potentially different temporal patterns of POs across different county-utilities, we utilized hierarchical GAMMs [54]. Such models allow for different temporal patterns for each county-utility. We used three different approaches to model seasonal patterns: day of the year (*DayInYear*) or minimum daily temperature for short-term seasonal trends, each with year as a categorical variable for long-term trends, and Julian Day (*JDay*, a continuous count of days) to model both seasonal and long-term trends. The inclusion of minimum daily temperature was intended to capture seasonality, potentially simplifying the model.

We used the Akaike's Information Criterion (AIC) to compare the models with different temporal patterns and chose the most parsimonious model when the AIC difference was less than two [60]. We verified whether spatial dependency was present by extending the best-fit models with a spatial smoother (Markov random field). Results indicated that there was no need to extend the models with spatial dependency.

**Model overview.** PO burden $SAIDI^*_{c,t}$ for a given county-utility service territory ($c$) and temporal or time-dependent variable ($t$) was modeled using a ZALN GAMM. This model consists of three steps:

1. Bernoulli Process (Probability of Zero Outage):

This component models the likelihood that there is no outage on a given day for a specific county-utility.

$$PO_{c,t} = Bernoulli(\pi_{c,t}) \tag{1}$$

The expected value of the probability of an outage absence can be expressed as:

$$E(PO_{c,t}) = \pi_{c,t} \tag{2}$$

With a log-odds representation:

$$Logit(p_{c,t}) = Intercept + Covariates_{c,t} + Dependency_{c,t} \tag{3}$$

The terms can be expanded for (see 'Exploring Model Structures' below for more information on the dependency terms):

$$Logit(p_{c,t}) = \beta_1 + f(\text{Poverty}_c) + f(\text{Unemployment}_c) + f(\text{Limited English}_c) +$$
$$f(\text{Temperature}_{c,t}) + f(\text{Wind Speed}_{c,t}) + f(\text{Precipitation}_{c,t}) + f_{County-utility}(\text{JDay}) + a_c \quad (3b)$$

Where $f(.)$ stands for a smoothing function, $\beta_1$ stands for the intercept, $f_{County\text{-}utility}$ is the smoother for *JDay* for each county-utility and $a_c$ is the smoother for random effects to model the county-utility specific intercept.

1. Log-Normal (LN) Process (Magnitude of Non-zero pOs):

When an outage occurs, this component models its magnitude or severity; The expected value of SAIDI on the original scale can be expressed as:

$$E(SAIDI_{c,t}) = e^{\mu_{c,t} + \frac{1}{2}\sigma_{c,t}^2} \quad (4)$$

Where μ is the mean of the natural log-transformed non-zero $SAIDI_{c,t}$ values.

$$\mu_{c,t} = Intercept + Covariates_{c,t} + Dependency_{c,t} \quad (5)$$

The terms can be expanded (see 'Exploring Model Structures' below for more information on the dependency terms:

$$\mu_{c,t} = \beta_1 + f(\text{Poverty}_c) + f(\text{Unemployment}_c) + f(\text{Limited English}_c) + f(\text{Temperature}_{c,t}) +$$
$$f(\text{Wind Speed}_{c,t}) + f(\text{Precipitation}_{c,t}) + f_7(\text{DayInYear}) + f_{Year}(\text{DayInYear}) + \quad (5b)$$
$$f_{County-utility}(\text{DayInYear}) + \beta_2 \times (Year) + a_c$$

Where $f(.)$ stands for a smoothing function, $\beta_1$ stands for the intercept, $\beta_2$ is the coefficient for year as a categorical variable, $f_{Year}$ is the smoother for the *DayInYear* for each year, $f_{county\text{-}utility}$ is the smoother for the *DayInYear* for each county-utility, and $a_c$ is the smoother for random effects to model the county-utility specific intercept.

2. Combining Parts 1 and 2:

The final model combines the Bernoulli and LN parts to provide a comprehensive representation of $SAIDI_{c,t}$ :

$$SAIDI_{c,t}^* \sim ZALN(p_{c,t}, \mu_{c,t}) \quad (6)$$

Where $SAIDI_{c,t}^*$ is the overall expected value of the SAIDI on the original scale and expressed with the following.

$$E(SAIDI_{c,t}^*) = (1 - \pi_{c,t}) \times e^{\mu_{c,t} + \frac{1}{2}\sigma_{c,t^2}} \quad (7)$$

**Exploring model structures.**   To investigate the driving factors of POs, we applied GAMMs that allowed for non-linear covariate effects of social vulnerability factors by using smoothing functions [54]. The GAMM software has the facility to determine whether a covariate effect is linear or non-linear (by estimating smoothing parameters) [54]. First, our models either included or excluded a global seasonal effect for each of the temporal variables: *DayInYear*, *JDay*, minimum temperature. Among the global seasonal models with the short-term

seasonal variables (*DayInYear* or temperature), we additionally allowed for the seasonal effects to differ by year. Second, we utilized the GAMMs to allow for temporal patterns that differed per county-utility. This is the smoothing equivalent of a random intercept and slope GLMM [54]. We either forced the county-specific temporal dependency to have the same smoothness for all county-utilities (shared smoothness) or allowed it to differ per county-utility (individual level smoothness) [54].

**Model fit.**   We assessed the goodness of fit for each stage of the zero-altered model and for the integrated ZALN model that combines these two stages using residual diagnosis based on scaled (quantile) residuals from the "DHARMa" package [61,62]. Model assumptions were verified by plotting scaled quantile residuals versus fitted values, versus each covariate in the model and versus each covariate not in the model. We found no major violations.

**Initial approaches.**   We first built generalized linear mixed effects models (GLMM) using the 'glmmTMB' package [63] and accounted for the hierarchical structure of the data by including random effects for the county-utility in all models. For the distribution, we began by using the Tweedie distribution (a special case of exponential dispersion models that can be used for positive, continuous, right skewed data with a point-mass at zero) [64]. We opted for the Tweedie distribution over a Gaussian (Normal) distribution because a Gaussian distribution could result in negative fitted values. However, these models were not able to cope with the many small values. We therefore applied zero-altered (hurdle) models with Gamma and negative-binomial distributions to the SAIDI data. However, the non-zero parts of the models resulted in a poor model fit, overpredicting small values. Additionally, model validation showed auto-correlated residuals. We therefore considered GLMMs with temporal auto-correlation terms, but due to over-fitting with an auto-correlation structure, we decided to apply GAMMs [54].

**Secondary analysis.**   We fit the same models that included a larger set of utilities that we excluded due to a lower baseline outage frequency and potential issues with missing not at random (MNAR).

## Results

### Validity of power outage data

During the study period, there were 117,890 unique POs among 14 utilities, with a median PO duration of 90.03 minutes (IQR, 41.07 to 182.68) for each customer affected. Statewide, our SAIDI estimates followed similar patterns to the EIA data, with the largest average duration of POs occurring in 2021 (S3 Table). Utility-level SAIDI values were also comparable (although some utilities deviated or were missing reliability data, S3 Table). Some county-utility territories had large variations in the natural log of SAIDI values from year to year, depending on extreme events such as wind, extreme rain, or even wildfire (Fig 2). Notably, Ferry county had large POs in 2020 at the time of one of the largest complex wildfires in Washington history in nearby Okanogan and Douglas counties [65,66].

We identified nearly all major events (defined as those affecting 50,000 customers for more than 1 hour) reported by utilities to the Department of Energy on DOE-417 [35]. We did not identify two major events: one affecting a large utility during a period when the API was offline, and another affecting Okanogan County that is not in the PowerOutage.us data (S4 Table). We identified one major event that was missing from the DOE database [35], and the dates, times, and county locations for major events in the DOE were sometimes incomplete or incorrect (S2 Fig). Notably, large POs often occurred in other counties at the same time as major events in the DOE database but may have not met the threshold for a major event for inclusion in the DOE database (e.g., Event 1, also affected Clallam and King Counties, Event 2

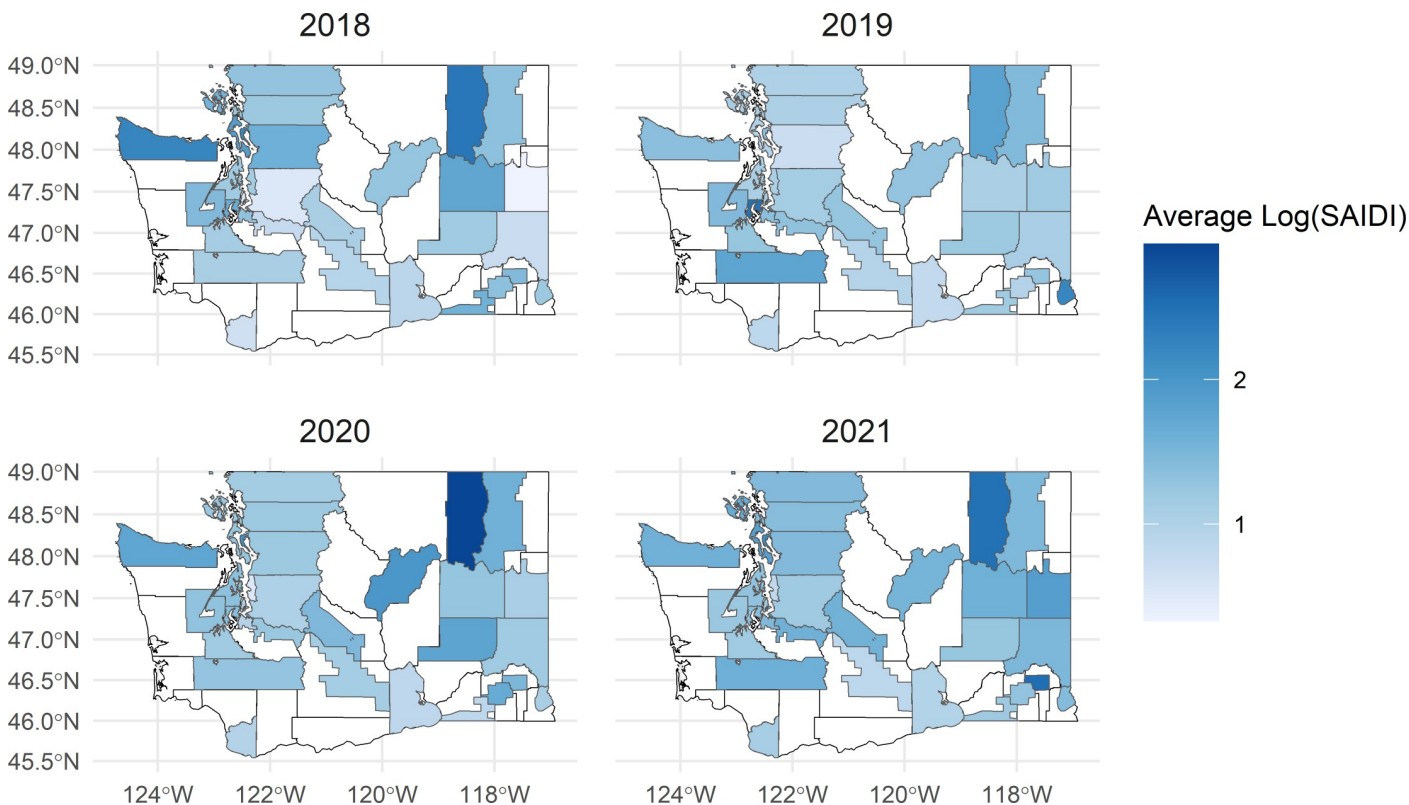

**Fig 2. Mean daily log of SAIDI values for each county-utility service territory (*n* = 31).** Areas shaded in white were not included in the PowerOutage.us data or were excluded from all analyses. Washington county boundaries were provided by the Washington State Department of Natural Resources [67]. Utility service territories provided by the Oak Ridge National Laboratory were modified based on maps provided on utility websites [46].

also affected King and Snohomish Counties). Certain areas experienced high SAIDI values (exceeding 60 minutes, see Table 1) during major events, even though these events did not qualify as major under the DOE definition (e.g., S2 Fig, Event 5, Ferry County).

In the primary analysis, there were 138 county-utility days with more than 10,000 customer POs and 9 county-utility days with at least 50,000 customer-POs (Table 1). Different definitions of major events or medically-relevant POs [13] resulted in widely different sample sizes and daily SAIDI values. Results for the secondary analysis were similar (S5 Table). Major event definitions such as POs affecting more than 50,000 customers excluded days with the highest SAIDI values.

We compared the downscaling estimation of county customer counts from census data as described by Do et al. [13] with estimation using utility-derived data. Two potential sources of error were identified in customer count estimates in prior research: first, the incorrect assumption that the ratio of meters to the total number of households and establishments remains constant regardless of the total number of meters; and second, the failure to adjust for incomplete utility coverage in the PowerOutage.us data. We show in S3 Fig that downscaling underestimates the number of customers in counties of smaller size (median percentage error: -12%, range: -37.5%-10.8%). This results in underestimating the number of customers for smaller counties. For instance, downscaling from census data estimated 3,526 customers in Ferry County, while utility-provided data indicated a higher count of 5,153 customers, resulting in a -32.2% error in the downscaled estimate. Additionally, considering the incomplete coverage of the PowerOutage.us data, we estimated the county's customer count at 1,828. Therefore, had

**Table 1. Description of medically relevant/major event definitions vs. non-zero outages.**

|  | Non-Zero Outages | $\geq 0.1\%$ for $\geq 8$ Hr[a] | Daily SAIDI $> T_{med}$[b] | $\geq$ 10,000 Max Affected | $\geq$ 50,000 Max Affected |
|---|---|---|---|---|---|
| **Sample Size, d** | 23,597 | 1,093 | 310 | 138 | 9 |
| **Daily SAIDI, min** *Median (Q1, Q3)* | 0.0 (0.0, 0.3) | 3.5 (1.4, 11.2) | 49.6 (30.6, 113.9) | 46.4 (17.0, 113.0) | 215.5 (82.5, 236.0) |
| *Range* | 0.0–1,432.0 | 0.1–1432.0 | 22.0–1432.0 | 2.1–1,178.3 | 66.6–427.5 |
| **Max Customer-Hr Affected** *Median (Q1, Q3)* | 24.3 (4.5, 142.8) | 371.5 (109.0, 1334.0) | 3,366.5 (1,009.3, 12,512.4) | 16,327.0 (12,551.8, 30,175.6) | 65,051.6 (51,122.1, 81,959.9) |
| *Range* | 0.0–154,908.8 | 1.2–44,864.0 | 58.0–154,908.8 | 10,031.6–154,908.8 | 50,074.2–154,908.8 |
| **Max Fraction of Customers Affected** *Median (Q1, Q3)* | 0.00 (0.00, 0.00) | 0.01 (0.00, 0.04) | 0.13 (0.08, 0.27) | 0.10 (0.06, 0.21) | 0.23 (0.11, 0.27) |
| *Range* | 0.00–1.00 | 0.00–1.00 | 0.02–1.00 | 0.02–1.00 | 0.09–0.42 |
| **Customer-Hr (Thousands)** *Median (Q1, Q3)* | 0.1 (0.0, 0.6) | 2.0 (0.5, 6.4) | 27.1 (5.5, 116.4) | 160.6 (72.6, 278.9) | 782.0 (769.5, 1286.4) |
| *Range* | 0.0–2272.3 | 0.0–596.4 | 0.5–2,272.3 | 16.4–2,272.3 | 630.5–2,272.3 |

$N$ = 31,714 d; Missing data: 808 (2.5%)

IQR: Interquartile Range

[a]POs of 8 consecutive hours or more could start and end on different calendar days; all days are included.

[b]$T_{med}$ was 21.93 minutes among all 23 county-utility territories.

we relied exclusively on the downscaled customer count, the error would have changed direction and increased to 92.4%.

## Data exploration

The highest variance inflation factor observed was for disability (VIF = 1.94, VIF = 1.69) in the primary and secondary analyses, respectively. The Pearson's correlation coefficients for all social vulnerability factors considered are in S6 Table.

## Model fit

We determined the best fit for modeling the seasonal effects. For the occurrence of POs, our optimal model allowed for individual temporal dependency by fitting county-utility-specific smoothers for Julian Day (*JDay*), with each allowed to have its own level of smoothness (Table 2). For the log-transformed average duration of POs, our optimal model featured a global smoother for seasonal effects (*DayInYear*) that was allowed to vary by year. This model allowed for individual temporal dependency by fitting county-utility-specific smoothers for the *DayInYear*, with each allowed to have its own level of smoothness. This means that while there is a general seasonal pattern, each county-utility can have its unique seasonal trend.

## Partial effects

The following are partial effects from the GAMMs for each covariate, accounting for all other covariates. The partial effects appear in the figures on two distinct scales: the log-odds scale for the absence of POs (Fig 3) and the natural logarithm scale for the daily average duration of POs (Fig 4). Notably, poverty, limited English proficiency, and unemployment had inconsistent relationships with PO burden. These variables had a non-linear relationship with PO frequency and no significant association with outage duration. There was, however, a nonsignificant trend of shorter outages for higher unemployment. Additionally, areas with the highest disability rates faced lengthier POs although there was no association with outage

**Table 2. Summary table of ZALN GAMM for SAIDI for the primary analysis.**

| | Binomial (Absence of Outage) | | Gaussian | |
|---|---|---|---|---|
| **Parametric Coefficients** | | | | |
| Component | Estimate | P-value | Estimate | P value |
| Intercept | -4.41 | < .001 | -0.53 | < .001 |
| Year[a] | | | | |
| 2019 | | | -0.17 | < .0001 |
| 2020 | | | -0.09 | 0.045 |
| 2021 | | | 0.01 | 0.729 |
| **Approximate Significance of Smooth Terms** | | | | |
| Component | edf | P-value | edf | P-value |
| s(Poverty) | 3.84 | < .0001 | 0.00 | 0.617 |
| s(Disability) | 0.50 | 0.122 | 1.89 | 0.113 |
| s(Unemployment) | 3.24 | < .0001 | 0.75 | 0.036 |
| s(Square Root of Limited English)[b] | 2.82 | < .0001 | 0.00 | 0.629 |
| s(Minimum Temperature) | 4.01 | < .0001 | 5.97 | < .0001 |
| s(Max Wind Speed) | 4.25 | < .0001 | 3.94 | < .0001 |
| s(Precipitation) | 1.00 | < .0001 | 3.95 | < .0001 |
| s(DayInYear): Year[c] | | | | |
| 2018 | | | 5.94 | < .0001 |
| 2019 | | | 4.00 | < .0001 |
| 2020 | | | 6.74 | < .0001 |
| 2021 | | | 6.36 | < .0001 |
| s(countyID) | 10.39 | < .001 | 19.08 | < .0001 |
| **Model Fit** | | | | |
| Component | Binomial | | Gaussian | |
| Deviance explained | .46 | | .17 | |
| N | 31,714 | | 23,597 | |

ZALN: zero altered log-normal; GAMM: generalized additive mixed model; SAIDI: system average interruption duration index; edf: effective degrees of freedom

Missing data: primary analysis, 808 (2.5%); secondary analysis, 3,968 (9.1%) county-utility days.

For brevity, the individual *JDay* and *DayInYear* smooths for each county-utility are not shown.

[a]Year reference category is 2018 for the Gaussian model and models including *JDay* do not include a categorical variable for Year.

[b]Indicator variable for limited English is transformed by taking the square root of its values.

[c]The variable to capture temporality and seasonality is *JDay* for the binomial model and *DayInYear* for the Gaussian model.

frequency, underscoring the potential complexity of these relationships. Thresholds for longer outages were relatively light to moderate, with spatially averaged maximum wind speeds over 8 m/s and daily precipitation accumulation over 32 mm being associated with longer PO durations, for example.

**Poverty.** The partial effect of the county-level percentage of the population living under the federal poverty level was non-linear, with lower probability outage absence (more likely to have a PO) for poverty levels above 14.68%, holding all other variables constant (at zero). Additionally, there was more frequent PO absence for lower poverty levels between 5.33% and 5.72% and between 10.59% and 12.73%. Poverty was not statistically significantly associated with PO duration at the $P < 0.05$ significance level.

**Unemployment.** Unemployment had a non-linear relationship with PO occurrence. The confidence intervals were wide and the smoother usually included zero. Counties with an unemployment rate of 4.27% to 5.45% were less likely to have a PO absence (more likely to

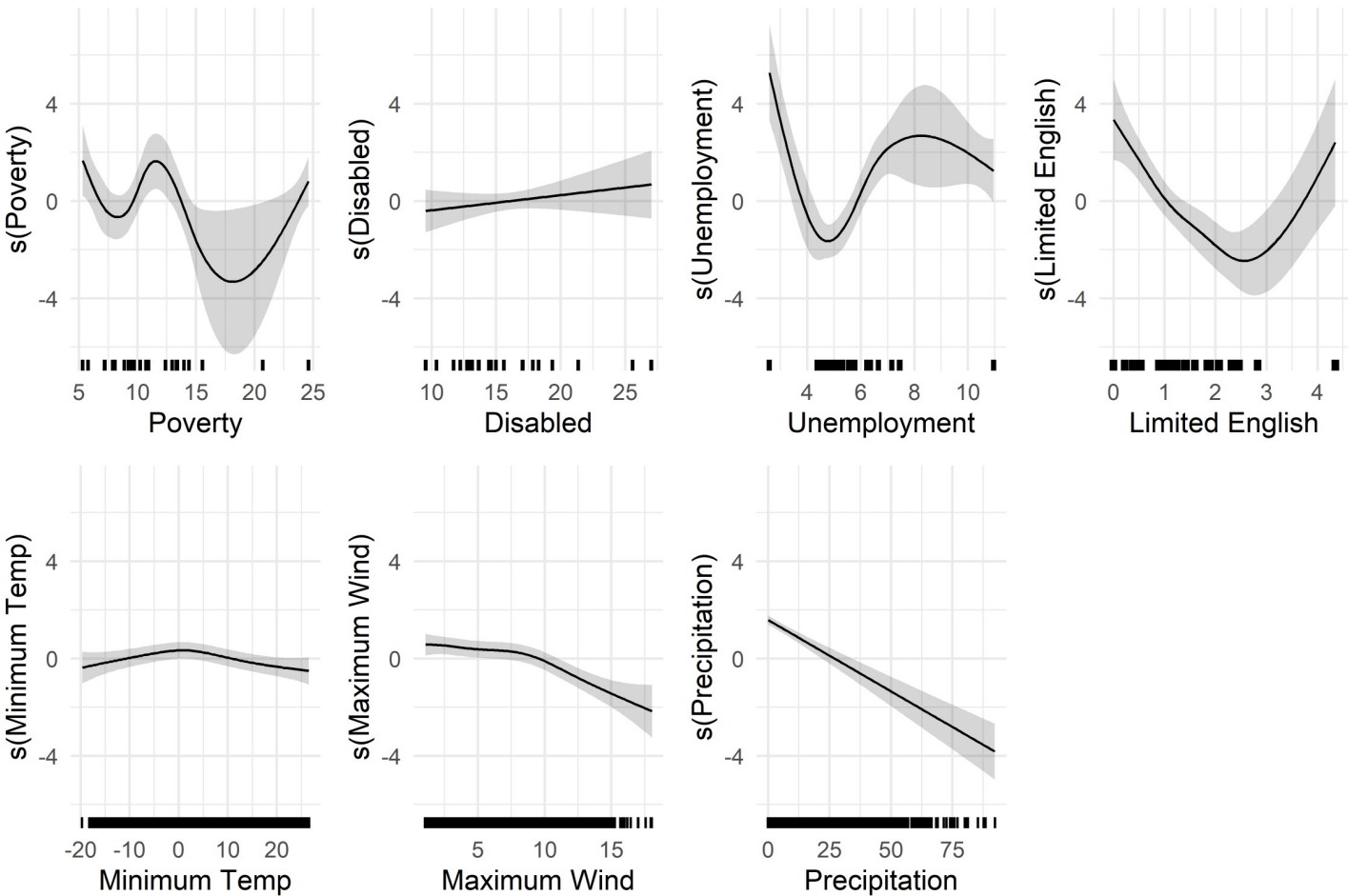

**Fig 3. Smooth effect on the log-odds of outage absence.** Partial effects from the fitted GAMM model predicting the log-odds of a power outage absence for 23 county-utility areas as a function of function of poverty (%), disability (%), square root of the % of limited English, unemployment (%), minimum temperature (°C), maximum wind (m/s), and precipitation (mm). The shaded areas represent the 95% confidence interval for the partial effects, the solid lines represent the smooth fitting curves of outage absence, and the x-axis represent the measured values of the explanatory variables. Rug marks along the x-axis represent data points from the original dataset ($n$ = 31,714) to indicate the distribution of observations.

have a PO), while those with an unemployment rate of 2.58%-3.34% and between 6.21% to 10.87% were more likely to have a PO absence (less likely to have a PO). There was a trend towards shorter outages for counties with higher unemployment rates, but it did not reach statistical significance. In summary, counties with low or high unemployment rates were less likely to have outages, and there was non-significant trend toward shorter unemployment for counties with higher unemployment.

**Disability.** The partial effect of county-level disability was not significantly associated with PO occurrence. The relationship between disability and PO duration was non-linear, with wide confidence intervals, and with longer POs for counties with over 23.14% of the adult civilian population with disabilities. Thus, counties with the largest percentages of the civilian adult population with disabilities had longer average POs.

**Limited english.** We transformed the percentage of households speaking limited English for analysis but have reverse-transformed them here for easier interpretation. The percentage of households speaking limited English also had a non-linear association with the probability PO absence. Counties with between 1.97%-9.73% of households speaking limited English were

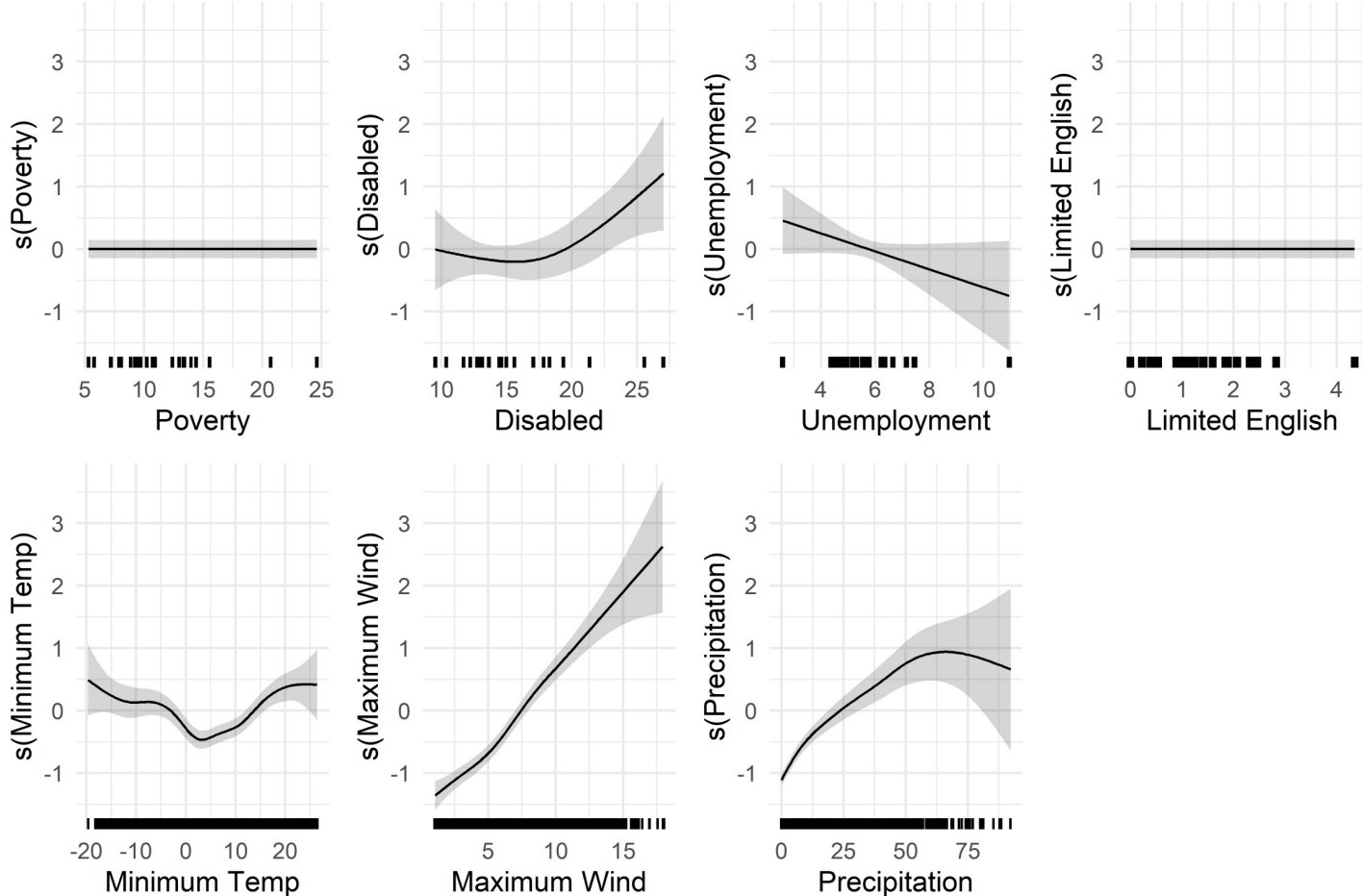

**Fig 4. Smooth effect on log of SAIDI in minutes.** Partial effects from the fitted GAMM predicting daily mean log-transformed SAIDI for 23 county-utility areas as a function of poverty (%), disability (%), square root of the % of limited English, unemployment (%), minimum temperature (˚C), maximum wind (m/s), precipitation (mm) for the effects of social vulnerability and weather on mean daily log-transformed SAIDI. The shaded areas represent the 95% confidence interval for the partial effects, the solid lines represent the smooth fitting curves of outage absence, and the x-axis represent the measured values of the explanatory variables. Rug marks along the x-axis represent data points from the original dataset (*n* = 23,597) to indicate the distribution of observations.

less likely to have an absence of POs (more likely to have a PO), holding all other covariates constant; confidence intervals were wider for counties with the highest rate of limited English proficiency. Those counties with under 0.55% of households speaking limited English were more likely to have a PO absence. There was no significant difference in duration at the *P* < 0.05 significance level.

**Weather.** In terms of weather, minimum temperature was not significantly associated with outage occurrence, while having a small increase in outage duration over 16.18˚C and decrease in outage duration between -1.07˚C and 11.52˚C. Both low and high temperatures had a wide confidence interval. Average maximum wind speeds over 11.18 m/s were associated with lower PO absence, while winds speed under 6.34 m/s were associated with increased PO absence. Wind speeds exceeding 8.05 m/s were associated with longer, and wind speeds under 6.86 m/s were associated with shorter PO duration. The average accumulation of precipitation exceeding 33.54 mm was associated with less PO absences and precipitation less than 21.43 mm was associated with PO absence. Daily precipitation exceeding 31.67 mm was associated with longer outage duration, while accumulation under 18.63 mm was associated with shorter

duration. There was greater uncertainty in outage duration for higher average maximum winds and precipitation.

**Seasonality.** The two parts of the ZALN were distinct in how they accounted for seasonality. The best fit model for presence/absence included a temporal variable of *JDay* and allowed for an individual effect of *JDay* for each county-utility. The model for log(SAIDI) included a global effect of *DayInYear* that was allowed to vary by year, and then allowed for individual effect of *DayInYear* for each county-utility. The timing of seasonal effects for the PO duration shifted each year; in two of the four years, winter had the largest seasonal effect, while in the other two years, late summer or early fall had the longer POs (Fig 5). County-utility areas had seasonal trends or temporal correlation for the models predicting absence (*n* = 6, 26%) and the average duration (*n* = 13, 57%). (Fig 6) shows that the partial effect of seasonality differs by county-utility areas.

## Secondary analysis

In the secondary analysis that included utilities with a higher number of days without observations, most results were similar. However, the best fit model for the occurrence of outages featured both a global smoother for seasonal effects (*JDay*) in addition to individual temporal

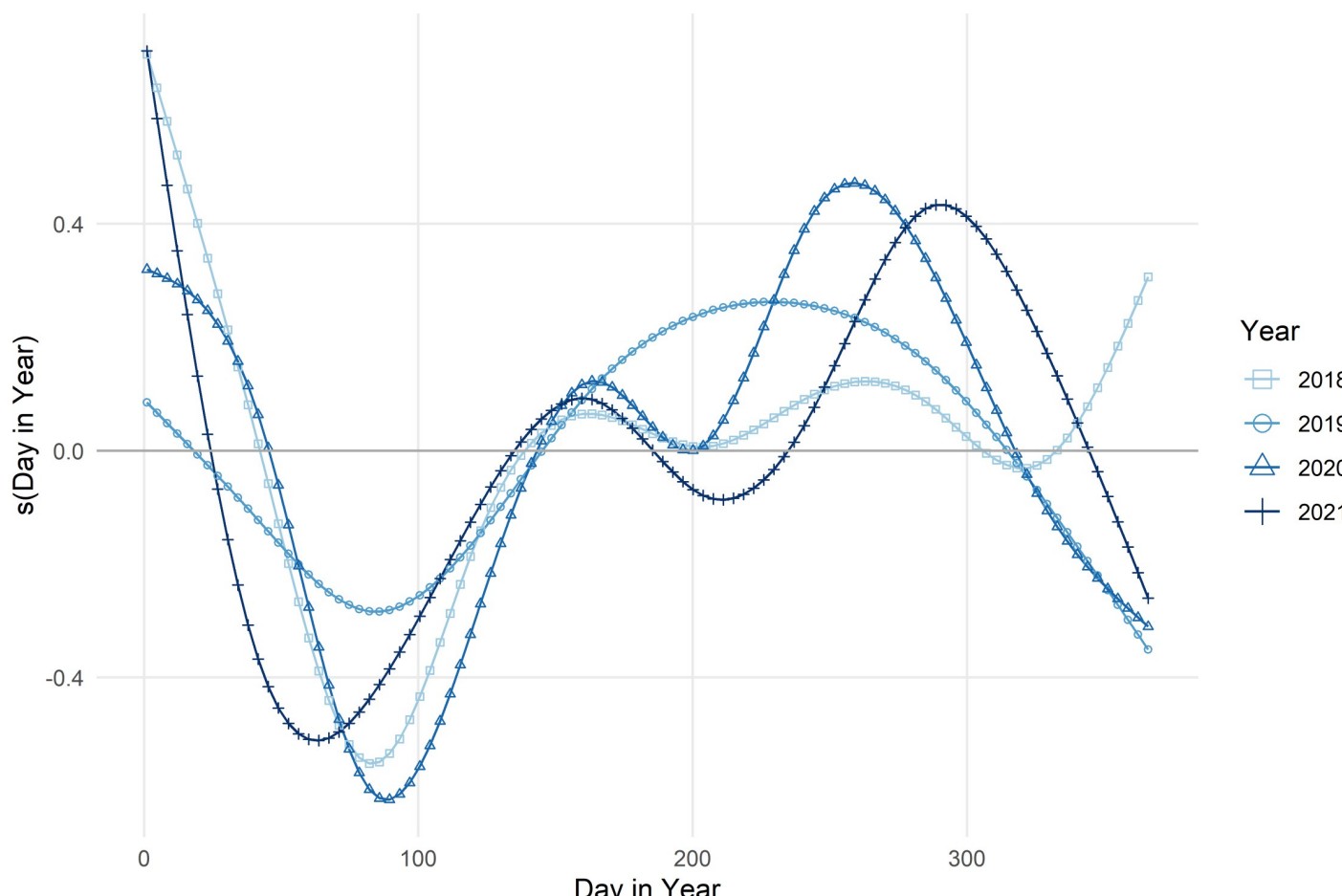

**Fig 5. Short-term seasonal effects on log of SAIDI in minutes for county-utility service areas.** Partial effects from the fitted GAMM predicting daily mean log-transformed SAIDI for each study year for 23 county-utilities and 23,604 county-utility days.

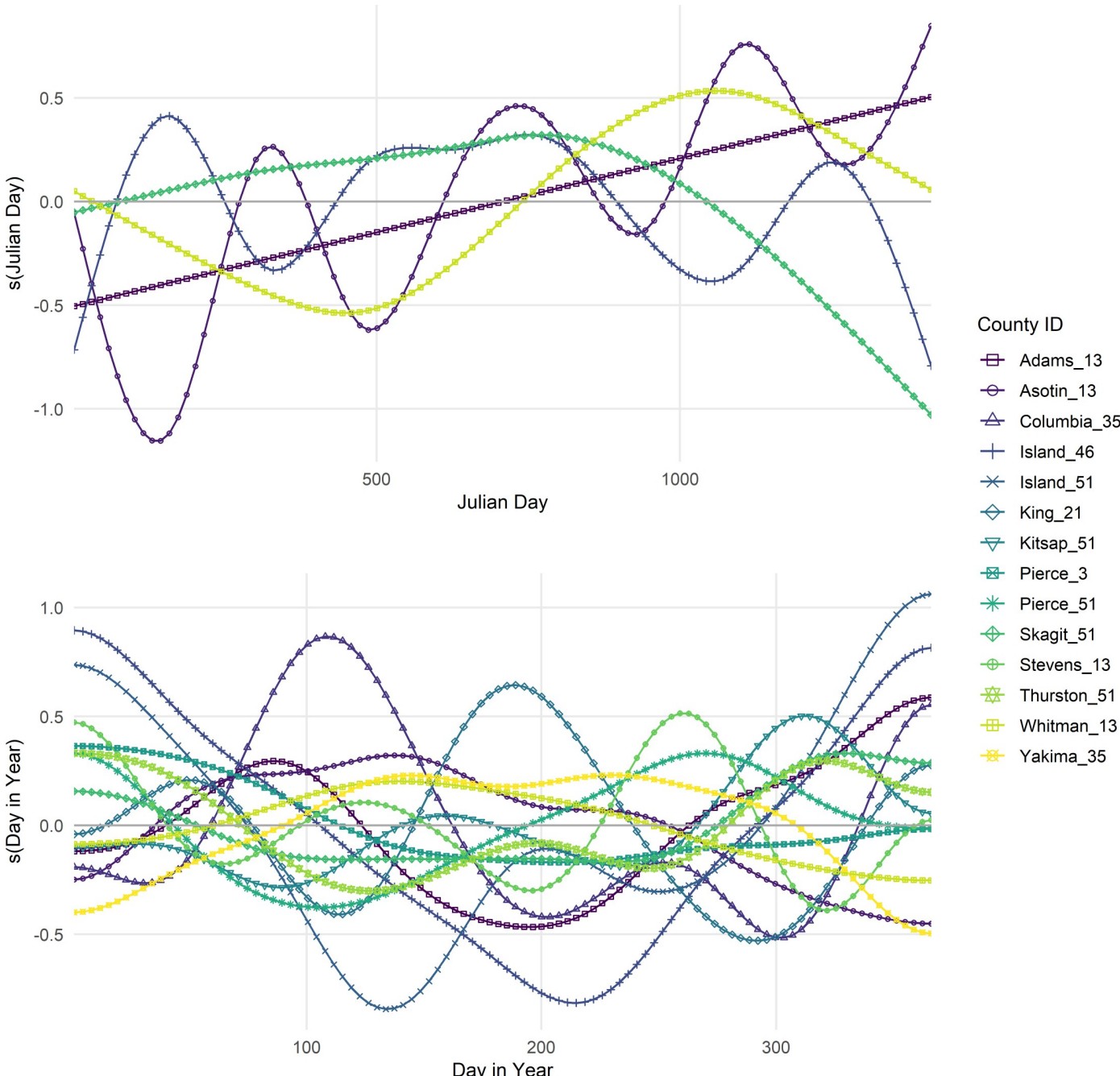

**Fig 6. Short-term seasonal effects on log-odds of outage absence for county-utility service areas.** The top panel includes the partial effects of seasonal effects for the county-utility ($n = 5$) on the presence/absence of outages. The bottom panel includes the partial effects of seasonal effects for the county-utility ($n = 13$) on the log(SAIDI). Figure includes only county-utility areas with 95% confidence intervals excluding zero.

dependency by fitting county-utility-specific smoother for *JDay* (S7 Table). Additionally, the most significant difference in results was for the partial effect of precipitation and poverty on the occurrence of POs (S4 Fig). Low and high precipitation resulted in higher absence of POs and the middle range of precipitation had a wider confidence interval, while the effect of poverty on PO occurrence was no longer statistically significant. Additionally, the partial effect of

disability on average PO duration was no longer statistically significant and the trend for unemployment disappeared (S5 Fig).

## Discussion

In our study, we conducted pre-processing of the PO data, described major PO events using absolute and relative definitions, and compared annual utility and state-wide utility metrics and major events identified using the PowerOutage.US data with federal datasets. We additionally examined the link between social vulnerability factors and PO burden in Washington State from 2018–2021. We modeled both covariates and response variables as continuous rather than dichotomous variables. We did so to avoid a loss of information and to avoid spurious threshold effects [37,38], whereby we find positive or negative effects only because of the choice of thresholds. Our analysis of daily SAIDI revealed an excess of zero values, non-linear patterns, missing data, and potential seasonal and temporal correlations. There were non-linear associations between social vulnerabilities and PO metrics, suggesting that the relationships are complex. The non-linear associations could be related to the level of analysis, in that urban and rural areas with varying physical and social vulnerabilities were aggregated. Our findings correspond with certain ecological studies [18], yet diverge from others [30,31], underscoring the difficulty in formulating consistent and generalizable insights from research on POs, especially given differing data sources, data quality, spatial resolution, pre-processing, and analytic choices.

Our findings generally agree with those of Mitsova et al. who researched county-level power restoration times following Hurricane Irma in Florida [18]. In their study, socioeconomic factors such as poverty and limited English proficiency were excluded from their final models due to a lack of statistical significance [18]. We found that there was a non-linear relationship for poverty and limited English proficiency with the log-odds of PO occurrence and no significant association with outage duration. In spatial lag models, the authors found longer restoration times in rural counties and counties with higher proportions of individuals with disabilities and Hispanic residents, and shorter restoration times for counties with higher unemployment [18]. We similarly noted longer outage durations for counties with higher proportions of individuals with disabilities and a non-significant trend of longer outage duration in counties with higher unemployment rates, and no significant association for outage frequency. The authors speculated that reduced outage duration in areas with higher unemployment could be attributed to residual confounding related to rurality [18]. Importantly, we were forced to exclude both rurality and population density from our models due to collinearity with social vulnerability factors. These variables could have captured distribution line density, factors that may have a causal relationship with PO burden. Other characteristics of rurality such as proximity to major urban areas may also affect restoration time. This highlights the challenge of distinguishing between physical and social vulnerability factors. Future work should consider examining urban and rural areas separately to better inform equitable resilience planning efforts.

Other research is conflicting with regards to the relationship between disabilities and power outages. In a study of the Winter Storm Uri's impact in Texas in February of 2021, Flores et al. examined the relationship of social vulnerability and major PO exposure, adjusting for urban/rural classification and population density [15]. In county-level analyses, higher percentages of Medicare populations using electricity-dependent DME consistently experienced fewer major outages [15]. These results differed from their findings in a non-representative survey that found individuals who used DME were more likely to experience major outages in the prior year [15]. Differences among research studies are difficult to explain, but it could be that there

are unmeasured factors, such as distance from critical infrastructure such as hospitals and differing priority in the power restoration hierarchy, or residual confounding related to distribution line density. The availability of more detailed and validated PO data could allow for more precise analyses that include a wider array of physical factors, such as the type of electrical infrastructure (above versus underground) [21], proximity to hospitals, customer distribution networks [14], or co-occurring hazards such as wildfire.

In contrast to our findings, some studies have suggested that lower socioeconomic status correlates with longer PO durations, though it is nuanced. In a cross-sectional study of localized POs for a single investor-owned utility between 2002–2003, Liévanos et al. implemented spatial error models to evaluate the relationship between a categorical variable of American Indian disadvantage and average POs (natural-log transformed) on the census block group level [14]. The authors found longer POs for areas with higher American Indian disadvantage and attributed these differences to bureaucratic decision-making rather than institutional bias [14]. Additionally, in a retrospective study of county-level power recovery following eight Atlantic hurricanes spanning 2017–2020, the authors suggested that socioeconomic vulnerability might affect PO duration, although significant associations were only confirmed for two of the eight storms [31]. This study also noted no significant correlations with other SVI themes such as household composition, minority status, or housing and transportation variables. In a cross-sectional study tracking power recovery for county subdivisions over eight months post-Hurricane Maria, Azad et al. utilized Quasi-Poisson models, considering both infrastructure (e.g., access to major roads) and socioeconomic indicators for county subdivisions [30]. The authors found that a 10% increase in poverty led to a 2% increase in recovery time but did not find any association for race or ethnicity. Physical factors such as distance to hurricane landfall, distance to major road arteries, landslides, and elevation were also critical factors.

We found that higher wind and precipitation resulted in more frequent and longer average POs. More extreme precipitation and increased severity and width of atmospheric rivers is expected in the Pacific Northwest due to progression of climate change [68,69], and may contribute to future POs. An important justice consideration is that despite the urgency to act on climate change, electric utilities and fossil fuel companies in the United States have established, managed, and funded interest groups to cast doubt on climate change and weaken climate policies [70,71]. They have also continued to expand fossil fuel infrastructure, despite evidence that new fossil fuel infrastructure is incompatible with limiting warming to 1.5˚C [70,72]. Electric utilities are not required to demonstrate that their activities- some of which are tied to climate change- do not contribute to the increase in POs. Furthermore, there is no mandate for reporting POs that disproportionately affect socially vulnerable groups, even though these groups are considered most vulnerable to climate impacts.

There is a need for validated PO data with finer patial resolution to allow for a better understanding of the impacts and distribution of POs. In this study, we identified numerous issues with PowerOutage.us data and conducted careful data processing treatments not previously described. To our knowledge, this is the first outage study to describe missing outage data as MNAR and to conduct separate analyses for more versus less reliable outage data. However, it is difficult to know how results of outage studies are affected by these data problems due to the lack of validated data to compare them with. We additionally demonstrated how customer count estimates could bias results. For example, downscaling from census counts could underestimate the number of customers in less populated counties, resulting in an overestimate of outage extent (proportion affected by outages). Our examination of the validity of PowerOutage.US data found problems with missing data in PowerOutage.US data and gaps and inconsistencies in federal datasets. DOE data on major POs [35] was frequently missing outage

durations and county locations for many major events, and the EIA data is missing some reliability metrics [36]. Moreover, the DOE definitions for major outages will primarily identify outages in urban areas and states and counties with large utilities. This is important because more populations reliant on electricity-dependent DME may be located in rural areas [73]. Recent research has identified four categories of PO events, based on size and recovery speed and can identify more moderate but meaningful outage events [74]. However, this approach still requires the use of thresholds. As of yet, it remains unknown what outage thresholds on a county or sub-county level have significance for public health and how those may change depending on other environmental hazards, such as extreme heat or wildfire smoke. Identification of these thresholds can allow for better public health surveillance and resource allocation and prioritization, but the lack of validated data could pose serious challenges to its use.

Although the Biden-Harris administration has encouraged utilities to standardize outage data sharing through the Outage Data Initiative Nationwide, participation is optional and detailed regional breakdowns are not compulsory [75,76]. Currently, a mere 125 (3.8%) of U.S. utilities share their outage data, highlighting a significant deficit in information [76]. For enhanced public health surveillance and accountability, there should be a requirement for electric utilities to report PO data and customer counts at more granular geographic levels, such as census tracts or block groups. Improved understanding of how PO burden is distributed according to the vulnerability of populations and co-occurring hazards could allow for infrastructure resilience planning and resources (e.g., solar with back-up batteries) to be appropriately allocated for prevention and mitigation of health impacts.

An important limitation of this study is that our county-level analysis and lack of inclusion of other physical factors may potentially obscure local disparities. Furthermore, our study's findings may not extend to other U.S. regions with greater deprivation or socioeconomic inequality, where county-level PO patterns could be more unevenly distributed. However, our study makes important methodological contributions, using a continuous PO metric for localized outages and raising questions about the quality of PO data and thresholds used in research. Employing a continuous PO metric allows for the detection of annual variability in the seasonal effects of POs, with these patterns also differing across county-utility regions. Such fluctuations could be indicative of seasonal influences or unidentified variables, such as wind gusts, wildfires, lightning, or annual shifts in utility operations and workforce. Despite the importance of temporal correlation and seasonal trends, other studies on differential PO exposure have analyzed data cross-sectionally and have not accounted for seasonal trends in their analyses [13]. Cross-sectional study designs [14,15,18,30,31,77] miss important seasonal data and do not capture the dynamic nature of POs, which can vary in intensity and affect different customers over time.

## Conclusions

Outage burden is an increasing public health threat due to the continued burning of fossil fuels and rising global temperatures, resulting in extreme weather. There is a low level of transparency into power outage exposure, with publicly available datasets possessing only crude temporal and patial resolution, with missing and sometimes incorrect data. A lack of customer counts within county or subcounty levels makes it difficult to accurately compare the outage probability or average duration across areas with different population sizes. Community organizations, scientists, regulators and policy makers lack sufficient information needed to judge whether outages are fairly or unfairly distributed among communities and to guide equitable planning efforts. Federal and state policy changes are needed to make these data more transparent and accessible.

## Supporting information

**S1 Fig. Number of subdivisions per utility each month.** Subdivisions are unknown when outage data is reported at the level of the county. Each panel corresponds to a utility, represented by an anonymized ID.
(TIF)

**S2 Fig. Major events 1–11.** Validation of PowerOutage.US data with the Department of Energy (DOE)-417, "Electric Emergency Incident and Disturbance Report." Outage events with dashed lines representing major events in the on the DOE-417, "Electric Emergency Incident and Disturbance Report." [1] Data representing the start and end date and time for the event according to the DOE files is demarcated with a blue dashed line. Areas with a blank x-axis indicate missing PowerOutage.US data. When impacted counties are missing from the DOE data, we assumed all counties in the utility service territory were affected.
(PDF)

**S3 Fig. Downscaling from census counts results in systematic error based on county size.**
(A) The ratio of downscaled census-based customer counts to census totals (households and establishments) versus households for Washington counties. (B) The ratio of utility-based customer estimates to census totals (households and establishments) versus the number of county households for Washington counties. (C) The ratio of downscaled to utility-based customer counts for the year 2021, with red point for Ferry County. (D) The ratio of downscaled to utility-based customer counts summed for utilities included in PowerOutage.us data for the year 2021, with red point for Ferry County.
(TIF)

**S4 Fig. Smooth effect on log odds of outage absence for secondary analysis.** Partial effects from the fitted GAMM model predicting the absence of a power outage for 31 county-utility areas as a function of function of poverty (%), disability (%), square root of the % of limited English, unemployment (%), rural (%), minimum temperature (˚C), maximum wind (m/s), and precipitation (mm). The shaded areas represent the 95% confidence intervals for the partial effects, the solid lines represent the smooth fitting curves of outage absence, and the x-axis represent the measured values of the explanatory variables. Rug marks along the x-axis represent data points from the original dataset ($n = 39,847$) to indicate the distribution of observations.
(TIF)

**S5 Fig. Smooth effect on log of SAIDI in minutes for secondary analysis.** Partial effects from the fitted GAMM predicting daily mean log-transformed SAIDI in Washington counties as a function of poverty (%), disability (%), square root of the % of limited English, unemployment (%), rural (%), minimum temperature (˚C), maximum wind (m/s), and precipitation (mm). The shaded areas represent the 95% confidence intervals for the partial effects, the solid lines represent the smooth fitting curves of log(SAIDI) and the x-axis represent the measured values of the explanatory variables. Rug marks along the x-axis represent data points from the original dataset ($n = 31,140$) to indicate the distribution of observations.
(TIF)

**S1 File. Data quality and supplementary references.**
(DOCX)

**S1 Table. Example of PowerOutage.** US data and Issues with Zero Values.
(DOCX)

**S2 Table. Data processing.**
(DOCX)

**S3 Table. Annual System Average Interruption Duration Index (SAIDI) for individual utilities and State: Study vs. EIA estimates from 2019–2021.** [a]Study data for 2018 was only a partial year and is not presented. [b]Statewide study data includes 15 utilities, while EIA data consists of all reporting utilities statewide. The EIA SAIDI values include major events from the EIA Electric Annual Power Report for Washington State. Utilities shaded in gray are included in the primary analysis, utilities shaded in white are additionally included in the secondary analysis. Empty rows indicate missing EIA data.
(DOCX)

**S4 Table. Major events reported to the Department of Energy (DOE) on the OE-417 "Electric Emergency Incident and Disturbance Report".**
(DOCX)

**S5 Table. Daily SAIDI and maximum fraction of customers out by major event definitions for the secondary analysis.** $n$ = 39,847 County-Utility Days. [a]Outages of 8 hours or more could start and end on different calendar days; all days are included. [b]$T_{med}$ was 22.12 minutes for all 31 county-utility territories.
(DOCX)

**S6 Table. Pearson's correlation for social vulnerability factors ($n$ = 31 county-utility areas, secondary analysis).** Shaded cells are those variables included in analyses. [a]Percent of Medicare Population; [b]Square root transformed. Poverty is defined as less than 100% of the federal poverty limit. BIPOC: Black Indigenous or Person of Color. DME: Electricity Dependent Durable Medical Equipment, Unemp: Unemployed civilian population.
(DOCX)

**S7 Table. Summary table of Generalized Additive Mixed Model (GAMM) for ZALN model of SAIDI (secondary analysis).** edf: effective degrees of freedom. Missing data: $n$ = 3,968 (9.1%) county-utility days. For brevity, we exclude the individual JDay and DayInYear smooths for each county-utility. [a]Year reference category is 2018 for the Gaussian model and models including JDay do not include a categorical variable for Year. [b]Indicator variable for limited English is transformed by taking the square root of its values. [c]The variable to capture temporality and seasonality is JDay for the binomial model and DayInYear for the Gaussian model. [d]The best fit model for the absence/presence model in the secondary analysis included a global term for seasonality (JDay).
(DOCX)

## Acknowledgments

We thank Drs. Tamara Odom-Maryon, Joan Casey, Janessa Graves, and Sterling McPherson in addition to Kim Zentz, Vivian Do, Heather McBrien, and Dmitri Kalashnikov for their advice at various stages of this project.

## Author Contributions

**Conceptualization:** Claire A. Richards, Solmaz Amiri, Von P. Walden, Julie Postma.

**Data curation:** Claire A. Richards, Alain F. Zuur.

**Formal analysis:** Claire A. Richards, Alain F. Zuur.

**Funding acquisition:** Claire A. Richards.

**Investigation:** Claire A. Richards, Von P. Walden.

**Methodology:** Claire A. Richards, Solmaz Amiri, Von P. Walden, Mohammad Heidari Kapourchali, Alain F. Zuur.

**Resources:** Claire A. Richards, Von P. Walden.

**Software:** Alain F. Zuur.

**Validation:** Claire A. Richards, Alain F. Zuur.

**Visualization:** Claire A. Richards, Alain F. Zuur.

**Writing – original draft:** Claire A. Richards, Von P. Walden, Alain F. Zuur.

**Writing – review & editing:** Claire A. Richards, Solmaz Amiri, Von P. Walden, Julie Postma, Mohammad Heidari Kapourchali, Alain F. Zuur.

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
