## [Decision Letter · Decision Letter 0]

23 May 2024

PONE-D-24-00148Association of Social Vulnerability Factors with Power Outage Burden in Washington State: 2018-2021PLOS ONE

Dear Dr. Richards,

Thank you for submitting your manuscript to PLOS ONE. After careful consideration, we feel that it has merit but does not fully meet PLOS ONE’s publication criteria as it currently stands. Therefore, we invite you to submit a revised version of the manuscript that addresses the points raised during the review process.

**ACADEMIC EDITOR:**
This work has merit. One reviewer provided major revisions and one reviewer provided minor revisions. Please address them appropriately. Specifically, it would be beneficial for the broader scientific community to have more clarity on the model formulation, notation, and use. 

We look forward to receiving your revised manuscript.

Kind regards,

Sudipta Chowdhury

Academic Editor

PLOS ONE

Journal Requirements:

2. Thank you for stating the following financial disclosure: "Washington State University New Faculty Seed Grant [PG00019865]."

4. We note that Figure 2 in your submission contain [map/satellite] images which may be copyrighted. All PLOS content is published under the Creative Commons Attribution License (CC BY 4.0), which means that the manuscript, images, and Supporting Information files will be freely available online, and any third party is permitted to access, download, copy, distribute, and use these materials in any way, even commercially, with proper attribution. For these reasons, we cannot publish previously copyrighted maps or satellite images created using proprietary data, such as Google software (Google Maps, Street View, and Earth). For more information, see our copyright guidelines: http://journals.plos.org/plosone/s/licenses-and-copyright.

Reviewers' comments:

Reviewer's Responses to Questions

**Comments to the Author**

1. Is the manuscript technically sound, and do the data support the conclusions?

Reviewer #1: Yes

Reviewer #2: Yes

2. Has the statistical analysis been performed appropriately and rigorously? 

Reviewer #1: Yes

Reviewer #2: I Don't Know

3. Have the authors made all data underlying the findings in their manuscript fully available?

Reviewer #1: Yes

Reviewer #2: Yes

4. Is the manuscript presented in an intelligible fashion and written in standard English?

Reviewer #1: Yes

Reviewer #2: No

5. Review Comments to the Author

Reviewer #1: This paper presents an analysis of the association between social vulnerability predictors, weather predictors, and power outage burden in Washington state. The major takeaways from this paper are that the relationship between social vulnerability factors and power outages are complex and nonlinear and that current research is limited by inconsistent and inaccurate outage data. The manuscript is well-written and technically sound. The authors perform rigorous statistical analyses and present their findings in a clear and detailed manner. The comparison between their approach of estimating customer counts and downscaling is helpful for future researchers in this area, as is their validation of PowerOutages.us datasets and DOE and EIA data. The discussion is particularly thorough and the authors did a great job of discussing their results in the context of relevant research. The paper could benefit from the addition of a conclusion section.

Revisions include:

- Please add a conclusion section to the manuscript.

- Line 101 contains an incomplete sentence. "scale of POs. is more sensitive...". Please fix this.

- Line 186: IEEE has already been defined in line 69 so there is no need to define it again. "IEE 1366-2005" should be changed to "IEEE 1366-2005". Please be more specific about the "other method" mentioned in line 187.

- Line 315, please define the acronym MNAR. Note that this is defined later on line 563.

- Remove the period from the end of the section titles on lines 264 and 454 to remain consistent with the rest of the manuscript.

Reviewer #2: This paper studies the relationships between power outages and social economic status of communities using data from Washington state. The paper consists of three aspects. First, the authors conduct data processing, with several more careful treatments that were not done by the prior work. Second, the paper proposes a model to relate power outages with social economic status of communities, i.e., Social Vulnerability Indices from CDC. Third, the paper uses the data to fit model parameters, and analyzes the results.

The strengths of the paper include careful and meticulous data processing. In particular, the authors separate major events from moderate disruptions using IEEE standards, and removing spurious data samples. Such efforts set the stage for data analysis. The model used appears to be different from those of the prior work. Overall, the study is relevant as Washington state suffers from frequent weather-induced power outages. Such a study has not been done in the prior work.

Weaknesses of the paper and suggestions include the following.

-First of all, the paper needs a major rewrite to be accessible by readers. For example, the paper can be structured to keep the key aspects of the study in the main text while moving the detailed implementation to the supplementary sections. For instance, the data section may keep what is needed for preprocessing and why that is important. How to do processing can be in the supplementary.

-The model equations, e.g., Equ (3) and (5) need to use standard and clear mathematical expressions. For example, what is the ``dependency” terms? Are covariate weighted linearly? Also, how to use the model also needs to be explained: are model parameters the same for all counties? Or the Bernoulli process pointwise, i.e., with different parameters for different counties? How are the model parameters estimated? Software used to implement the model can be moved to the supplementary.

-Analysis of the results is now in ``discussion” section, which seems to be too long to be clear. To highlight the findings, the results and analysis can be a separate section. It needs to be made clear what key results are obtained, and how the results differ from the prior work (and why if known). For example, in what way, the outage durations are non-linear in SVI? Also, which part of the extensive data processing makes the difference to the results? Why are the impact of outages inconsistent in terms of SVIs?

Some detailed comments:

-It will be more clear to specify the actual vertical axis in Figs 3, 4, 5, 6 (rather than ``partial effects”).

-The paper uses DoE criterion for major events, i.e., outages that affected more than 50,000 (or >10,000) customers. Such a criterion is for transmission grid. For distribution grid, separating major events from moderate ones are considered in a prior work below.

Reference: Afsharinejad, A., et al., “Large-scale data analytics for resilient recovery services from power failures,” Joule Cell Press, 2021

Overall, this reviewer is positive about the paper but thought a revision is needed.

6. PLOS authors have the option to publish the peer review history of their article (what does this mean?). If published, this will include your full peer review and any attached files.

Reviewer #1: **Yes: **Jesse Dugan

Reviewer #2: No

---

## [Author Response · Author response to Decision Letter 0]

15 Jun 2024

Please see the response to reviewers letter attached.

---

## [Decision Letter · Decision Letter 1]

11 Jul 2024

Association of Social Vulnerability Factors with Power Outage Burden in Washington State: 2018-2021

PONE-D-24-00148R1

Dear Dr. Richards,

We’re pleased to inform you that your manuscript has been judged scientifically suitable for publication and will be formally accepted for publication once it meets all outstanding technical requirements.

Kind regards,

Sudipta Chowdhury

Academic Editor

PLOS ONE

Additional Editor Comments (optional):

Reviewers' comments:

Reviewer's Responses to Questions

**Comments to the Author**

1. If the authors have adequately addressed your comments raised in a previous round of review and you feel that this manuscript is now acceptable for publication, you may indicate that here to bypass the “Comments to the Author” section, enter your conflict of interest statement in the “Confidential to Editor” section, and submit your "Accept" recommendation.

Reviewer #1: All comments have been addressed

Reviewer #2: All comments have been addressed

2. Is the manuscript technically sound, and do the data support the conclusions?

Reviewer #1: (No Response)

Reviewer #2: Yes

3. Has the statistical analysis been performed appropriately and rigorously? 

Reviewer #1: (No Response)

Reviewer #2: Yes

4. Have the authors made all data underlying the findings in their manuscript fully available?

Reviewer #1: (No Response)

Reviewer #2: Yes

5. Is the manuscript presented in an intelligible fashion and written in standard English?

Reviewer #1: (No Response)

Reviewer #2: Yes

6. Review Comments to the Author

Reviewer #1: (No Response)

Reviewer #2: The authors have addressed most of the questions raised by the reviewer. There are a certain minor issues remaining such as the format of the equations, and labels of the figures. These do not prevent the acceptance of the publication.

7. PLOS authors have the option to publish the peer review history of their article (what does this mean?). If published, this will include your full peer review and any attached files.

Reviewer #1: No

Reviewer #2: No

---

## [Editor Report · Acceptance letter]

18 Jul 2024

PONE-D-24-00148R1 

PLOS ONE

Dear Dr. Richards, 

I'm pleased to inform you that your manuscript has been deemed suitable for publication in PLOS ONE. Congratulations! Your manuscript is now being handed over to our production team.

Kind regards, 

on behalf of

Dr. Sudipta Chowdhury 

Academic Editor

PLOS ONE